# A Phenotype-Based Approach for the Substrate Water Status Forecast of Greenhouse Netted Muskmelon

**DOI:** 10.3390/s19122673

**Published:** 2019-06-13

**Authors:** Liying Chang, Yilu Yin, Jialin Xiang, Qian Liu, Daren Li, Danfeng Huang

**Affiliations:** School of Agriculture and Biology, Shanghai Jiao Tong University, Shanghai 200240, China; Changly@sjtu.edu.cn (L.C.); yyl0926@sjtu.edu.cn (Y.Y.); jocelyn_xjl@alumni.sjtu.edu.cn (J.X.); liuqiansmd@sjtu.edu.cn (Q.L.); chhblidaren@sjtu.edu.cn (D.L.)

**Keywords:** muskmelon, phenotype, random forest algorithm, cultivation substrate water status, forecasting

## Abstract

Cultivation substrate water status is of great importance to the production of netted muskmelon (*Cucumis melo* L. var. reticulatus Naud.). A prediction model for the substrate water status would be beneficial in irrigation schedule guidance. In this study, the machine learning random forest model was used to forecast plant substrate water status given the phenotypic traits throughout the muskmelon growing season. Here, two varieties of netted muskmelon, “Wanglu” and “Arus”, were planted in a greenhouse under four substrate water treatments and their phenotypic traits were measured by taking the images within the visible and near-infrared spectrums, respectively. Results showed that a simplified model outperformed the original model in forecasting speed, while it only uses the top five most significant contribution traits. The forecast accuracy reached up to 77.60%, 94.37%, and 90.01% for seedling, vine elongation, and fruit growth stages, respectively. Combining the imaging phenotypic traits and machine learning technique would provide a robust forecast of water status around the plant root zones.

## 1. Introduction

Netted muskmelon (*Cucumis melo* L. var. reticulatus Naud.) is a popular fruit in the fresh market due to its unique netted patterns, esthetically pleasing shape, high sugar content, and good taste, which makes netted muskmelon a widely planted fruit among growers worldwide. Currently, there are more than 460,900 hectares of netted muskmelon planted each year [1]. 

Growing netted muskmelon requires strict cultivation conditions and complex management strategies, particularly for the water status in the cultivation substrate [2,3]. Inadequate substrate water status would lead to yield loss, poor quality, low sugar content, and unaesthetic netted patterns [4,5,6]. Dogan et al. [7] discovered that cultivation substrate water content serves as a critical factor influencing sugar accumulation and specifically netted patterns formation. Therefore, it is necessary to regulate and control the substrate water content throughout all growing stages of netted muskmelon for quality improvement considerations.

Root zone water status is the moisture situation of plant root and is highly correlated to plant growing status [8]. Monitoring the substrate water status of plant root zone could help to establish an effective management strategy of precise irrigation in crop production. Several commercialized soil moisture sensors, such as tensiometer, neutron probes, and time domain reflectometry probes have been widely used in pot planting but they are limited in detecting the substrate water status because of the high cost and unreliable data collection. Furthermore, one probe only reflects the substrate water content of a single point rather than the water statuses of different pots [9,10,11]. 

With the development of sensing technology, real-time monitoring of physiological and ecological information of the plant itself provides an approach to indicate root zone water status [12]. Many studies have showed that the phenotypic traits of plant can reflect the water content of plants [13,14,15,16]. Phenotypic traits such as leaf area, leaf angle, and thermal temperature show strong correlations with the root zone water status [17]. Moreover, with the development of plant phenotypic monitoring technologies, now it is viable to carry out automatic high-throughput monitoring without damaging the plants, thus quickly acquiring the phenotypic traits and remarkably shortening the test cycle [18,19,20]. However, most of the plant-based irrigation methods are still at research/developing stage and little used yet for practice (except for thermal sensing in some situations). Systematical research on phenotypic traits selection for irrigation has seldom been reported.

A rapid and precise data analysis method is needed for plant-based information acquisition. The application of machine learning algorithm could be the solution. It is by easy data preparation, low computation complexity, fewer input factors, and availability to handle uncorrelated characteristics provides a solution for higher model forecast accuracy and better model adaptability [21]. Therefore, this algorithm can meet the need of high-throughput real-time monitoring. At present, many modern agricultural management technologies, such as agricultural internet of things (IOT) decision-making system [22], agricultural information construction, identification of crop diseases [23], and yield forecast [24], etc., have been created in aid with machine learning algorithm. Many studies have reported plant stress phenotyping using machine learning and prediction of plant water status, but few have focused on phenotyping root zone water stress [25,26]. It is crucial to detect plant water stress at the early stage before any damage observed with visible wilting, which affect subsequent plant growth and quality. Correlating root zone water status with plant phenotype to supervise irrigation system could be a practical way in precision irrigation system with easy operation and quick response. However, machine learning modeling in analyzing the phenotypic trait of greenhouse crops and judging their root zone water environment has seldom been used.

In this study, we obtained the phenotypic traits of greenhouse netted muskmelon by high-throughput phenotypic monitoring technology and built a classification forecast model for the water status of the netted muskmelon cultivation substrate using the random forest algorithm. The substrate water status of netted muskmelon at different growth stages can be determined in time, which offers a novel solution for the real-time irrigation of muskmelon.

## 2. Materials and Methods

### 2.1. Materials and Treatments

Experiment I was conducted in the greenhouse of Shanghai Zealquest Technology Co., Ltd, China (GPS coordinate: 31°11′N, 121°36′E) from April to July in 2016. The experiment II as model validation was conducted in a multiplan greenhouse at Shanghai Jiaotong University, China (31°11´ N, 121°36´E) from August to November in 2016, and the glasshouse heating, ventilation, and internal and external shading were all automatically regulated by Priva Software (Priva Company, Zuid Holland, the Netherlands) in these two experiments. Two cultivars of netted muskmelon, “Wanglu” with dense nettings and “Arus” with sparse nettings, were used in these two experiments. The muskmelon seedlings were first cultivated in plug at greenhouse, and then transplanted to pots at two-leaf and one-half stage. The muskmelons were vertically planted in pots with a total volume of 16 L of substrate containing 2:2:2:1 mixture of meteorite/perlite/peat/organic fertilizer. The bulk density and saturated water content were 0.21 g/mL and 140%, respectively.

For considering the feasibility of moisture level control and plant growth after transplanting, four relative water content levels were applied to these two experiments. The four water treatments with alternating 20%~35% relative water content (RWC), 35%~50% RWC, 50%~60% RWC, and 60%~70% RWC at seedling stage, 30%~40% RWC, 40%~50% RWC, 55%~70% RWC, and 70%~85% RWC at vine elongation stage, and 35%~45% RWC, 45%~55% RWC, 55%~65% RWC, and 65%~80% RWC at fruit development stage (Figure 1). Substrate water status was measured by the relative water content that was the percent of volumetric water content comparing to field water capacity. Due to the dynamic change of root zone moisture status, an upper limit was set for the moisture level of root zone. In other words, irrigation was stopped when the root zone moisture reached the upper limit. In this study, a Soil-watch Multi-parameter Monitoring System (Soil-Watch, Washington, USA) was used to measure the relative water content of root zone (%). Five plants of each cultivar were randomly assigned to every treatment. Irrigation was conducted manually for all plants once a day according to the monitoring system.

### 2.2. Image Acquisition and Trait Extraction

A commercial phenotyping system (Scanalyzer 3D, LemnaTec GmbH, Würselen, Germany) was used for image acquisition (Figure 2), and imaging was performed every three days during the whole growth period for experiment I and every week for experiment II. At each imaging day, a total of 40 potted plants were placed at a fixed position under certain lighting conditions in an imaging room, where the light background was consistent and obtained stable color values, and two types of image were acquired simultaneously. Phenotypic images taken within the visible and near-infrared spectrums (Figure 3) were collected from both the top and sides of these plants. A total of 2160 images were collected during the experiment I and 960 images during the experiment II. 

Phenotypic image processing steps (Figure 4) and image processing procedure in LemnaGrid consisted of four main steps shown as the flowchart in Figure 5: (1) image preprocessing, extracting target images from LemnaBase; (2) segmentation, separating target plant from the background in the image; (3) feature extraction, analyzing segmentation result and producing phenotypic traits; and (4) post-processing, summarizing feature extraction results of all target images and exporting as “.xls” file.

### 2.3. Model Development of Substrate Water Status

The development of substrate water status discrimination was consisted of three steps: (1) data preprocessing, data cleaning, data conversion, organizing data for phenotypic analysis; (2) phenotypic traits screening, removing those phenotypic traits with insignificant difference between treatments of substrate moisture level; (3) model development, using random forest modeling algorithms to classify substrate water status. MetaboAnalyst statistical analysis module (http://www.metaboanalyst.ca) was implemented for data preprocessing and phenotypic traits screening, and R language (windows, 3.3.3 version, R core Team, 2016) was implemented for model development.

### 2.4. Data Preprocessing

In MetaboAnalyst statistical analysis module, data of extracted phenotypic traits from each image were organized into one datasheet by sample number (row) and phenotypic trait (column). Columns with more than 50% empty cells were removed. In the case of columns with data missing but less than 50% empty cells, the value of these cells were replaced by with a small value (the half of the minimum positive values in the original data). All the trait values were normalized using an autoscaling method (mean-centered and divided by the standard deviation of each variable) [12].

### 2.5. Phenotypic Traits Screening

Among those phenotypic traits extracted by imaging processing, traits which are insignificantly relevant to treatments may influence the modeling accuracy. In this study, one-way analysis of variance (ANOVA) was applied to screen traits resulting an optimal set of explanatory variables for discrimination model development. Traits with P > 0.05 (Fisher’s LSD method) were excluded from the traits set. In addition, the result of ANOVA presented a preliminary overview of the significance of each trait to treatments [12].

### 2.6. Model Development and Forecast Ability Assessment

Random forest is an ensemble classifier that consists of many decision trees and outputs the class that is the mode of the class’s output by individual trees. Random forest algorithm was developed by Leo Breiman [27], boasting the advantages in big data processing of excellent performance in datasets, resistance to overfitting, good noise immunity, and strong adaptability to datasets, fast training speed, easy realization, among others. 

A random forest (RF) algorithm has two parameters, the number of decision-making trees (ntree) and the number of features that are used to find the best features (Mtry). In the experiment, Mtry = 1, 2, 5, 8 and ntree = 500 were tested for benchmarking the RF models with the highest prediction accuracy. The random forest package of R language [28] was used for modeling, and the model adopted the 10-fold cross validation that has been repeated for three times for the training. 

The total samples of experiment I were selected as the training dataset for model development and optimization and the experiment II as the test dataset. The data processing was used the IBM SPSS statistics 20 and the self-compiled script in R language version 3.3.3 (R Core Team). Data mining and standardization were implemented with the statistical and analytical modules of MetaboAnalyst 3.0 software(https://www.metaboanalyst.ca/). ConfusionMatrix. Train function was used to assess forecast ability of the random forest model, and the mean value of forecast accuracy for cultivation substrate water content for all growing stages was obtained. 

## 3. Results

### 3.1. Subsection

#### 3.1.1. Extraction of Phenotypic Traits

There were 2160 images captured at different growth stages throughout the experiment, and 29 traits of three categories were extracted (Figure 3, Figure 4 and Figure 5 and Table 1). These traits include 9 morphological traits and 6 color traits, 14 near-infrared feature traits, which were acquired from visible and near-infrared spectrums, respectively.

#### 3.1.2. Analysis of Phenotypic Traits

Phenotypic data were analyzed by one-way analysis of variance (ANOVA). As shown in Table 1, at the seedling stage, 9 traits (diameter, compactness, eccentricity, mean blue, mean color blue variance, and mean color red variance, A1(Number of plant pixels with NIR intensity in 122–138), R1(Ratio of plant pixels with NIR intensity in 122–138) and R3(Ratio of plant pixels with NIR intensity in 170–186) showed no significant differences among four water treatments. At the fruit development stage, only two traits, mean color green and R7(Ratio of plant pixels with NIR intensity in 234–250), had no significant difference. At the vine elongation stage, all traits showed significant differences, and three traits, B6(Ratio of plant pixels with NIR intensity in 218–234), B3 (Ratio of plant pixels with NIR intensity in 170–186) and diameter, had the largest F-value among all treatments.

#### 3.1.3. Machine Learning Modeling and Forecast Ability Assessment of Phenotypic Traits

Further, the traits showing significant difference (P < 0.05) were standardized and imported to machine learning model, which included 18 traits at the seedling stage, 29 traits at the vine elongation stage, and 27 traits at the fruit development stage. Then, a random forest classification model was created to forecast cultivation substrate water content (Table 2). The forecasting accuracy from traits at seedling stage, vine elongation stage, and fruit development stage were 78.5%, 95.7%, and 99.5%, respectively. Among three growing stages of muskmelon, forecasting accuracy of random forest model at fruit development stage was the highest, while it took the longest forecast time costing 136.16 s due to input of a large number of traits.

Confusion matrix was implemented to validate the developed models, and the average forecast accuracies of cultivation substrate water contents for all growing stages were obtained (Table 3). Among three growing stages of muskmelon, the lowest forecast accuracy of substrate water content was at seedling stage given the phenotypic traits, which was 78.48%, while the highest forecast accuracy was obtained at fruit development stage based on the phenotypic trait data, which reached up to 99.55%. Also, the forecast accuracy based on traits at vine elongation stage reached up to 95.74%. All the accuracies from three growing stages could meet the needs of predicting substrate water content. The lowest forecast accuracy at seedling stage might be due to the less traits with significant difference among four water treatments.

#### 3.1.4. Trait Contribution Analysis and Model Simplification

In order to check the possibility of reducing phenotyping cost and model training time but without substantially decreasing the prediction accuracy, a sensitivity analysis was conducted to assess the contribution of each trait in the developed models. A simplified model with less explanatory variables was developed based on the contribution evaluation of traits for the model implemented in substrate water status detection. At the seeding stage of muskmelon, the top five most significant contribution traits for random forest modeling were the projection area, the number of plant pixels with near-infrared (NIR) intensity in 170–186 (A3), the number of plant pixels with NIR intensity in 186–202 (A4), the number of plant pixels with NIR intensity in 202–218 (A5), and the ratio of plant pixels with NIR intensity in 218–234 (R6). The top five most significant contribution traits at vine elongation stage refer to the minimum height of plant projection (min height), the convex hull circumference, the circumscribed circle diameter of plant projection (diameter), the number of plant pixels with NIR intensity in 122–138 (A1), and the ratio of plant pixels with NIR intensity in 170–186 (R3). However, the top five most significant contribution traits were all morphology traits at fruit development stage, and they were the minimum width of plant projection (min height), the minimum external polygonal area of plant projection (min EP projection area), the convex hull circumference, the circumscribed circle diameter of plant projection (diameter), and the minimum external rectangle area of plant projection (min rectangle area) (Table 1). Thus, the top five traits in terms of contribution values were used in the simplified random forest modeling (Table 1), and they are varied among different growth stage models. Although the model was simplified, the forecasting accuracy of substrate water content remains high value for all the three growing stages of muskmelon.

Confusion matrix was implemented to validate the simplified random forest model. Results showed that the mean value of forecasting accuracy for cultivation substrate water content decreased by 0.90%, 1.30%, and 9.50% at seedling stage, vine elongation stage, and fruit development stage, respectively (Table 3). Although the forecasting accuracy had a little decline, it still meets the needs for forecasting of substrate water content in real practice.

## 4. Discussion

This study developed a forecast method of substrate water status during plant growth by integrating phenotyping and machine learning techniques, and using muskmelon plants. Different phenotyping traits were from three growth stages and two muskmelon cultivars, three types of phenotyping traits, such as morphological traits, color traits, and near-infrared traits, were acquired as the inputs for the model. The model had high accuracy at three different stages, with the highest value as 94.37% at vine elongation stage, but the contribution of phenotyping traits to the model were varied among the three growth stages.

At the seedling stage of muskmelon, almost all the traits that are sensitive to water status and have a significant contribution value to random forest model are related to image-gray-level and can be acquired by near-infrared monitoring. Therefore, it can be concluded that under all water treatments, the morphology differences among muskmelons are too minor to be easily captured by the sensor, while the difference in image-gray-level under near-infrared monitoring was significant. This result is consistent with the findings by Chen et al. [29] and Story and Kacira [21] that the changes of leaf cell demonstrate the powerful reflectivity within the near-infrared region, and are consistent with the research results by Yang et al. [30] and Buddenbaum et al. [31] that indicated the NIR has the potential to provide for rapid, non-destructive assessment of the leaf water potential in plant seedlings. Therefore, forecasting of the plant substrate water status can be implemented through the near-infrared image data at seedling stage of muskmelon, but cannot be implemented at the other two growth stages. 

The traits that were sensitive to the substrate water status and contributing a significant value to the random forest model are nearly all morphology traits at fruit development stage. This indicates that water has a significant effect on the morphology traits of melon, whose morphology traits contribute greatly to the random forest model. Therefore, it is possible to forecast the substrate water status by using the morphology traits at the fruit development stage. At the vine elongation stage, most of the sensitive traits refer to morphology and near-infrared spectrum traits. This result is consistent with the findings reported in the area of spectroscopy and remote sensing imaging, such as Möller et al. [28], Veysi et al. [32], Shivers et al. [33], Chen et al. [34], and Chen et al. [35], that the water stress and canopy water status of field crop can be effectively monitoring using optical imaging technology.

This study demonstrates the potential of the machine learning approach in forecasting the substrate water status based on complex plant phenotyping traits. This method could promote the plant-based irrigation decision-making and implementation in practice. Therefore, the accumulative effect of water stress on plant growth during the growing period should be minimized with real-time discrimination to control the potential yield loss. Our presented work is capable of serving as a basis and supporting for intelligent greenhouse management, especially for irrigation management. Population phenotyping and water status discrimination are also worth further study due to plant canopy mutual occlusion and growth competition. In this study, plant images and other phenotypic traits were taken under controlled light and background colors to ensure prediction consistency. In real practice, however, the lighting condition and background color are not as consistent, which impose severe interferences for the image acquisition consistency. Therefore, further research with on-site traits collection is needed.

## 5. Conclusions

In this study, the cultivation substrate water status of greenhouse muskmelon at different growing stages were predicted based on melon morphology, color, and near-infrared feature traits as the input parameters for the random forest classification model. Prediction reached high speeds of 12.88, 17.44, and 11.14 s and high accuracies of 77.60%, 94.36%, and 90.01% at the seedling, vine elongation, and fruit development stages, respectively. The melon root rhizosphere water status of substrate can be forecasted through the minor changes of muskmelon phenotypic trait, thus providing a solution to the limited probe sensors monitoring area, instable data, and weak instantaneity. Therefore, phenotypic imaging combining machine learning modeling can effectively forecast the substrate water status of muskmelon at different growing stages. This demonstrates that monitoring plant phenotypic traits can be used to determine substrate water status, and the forecast of substrate water status can be implemented with random forest model.

## Figures and Tables

**Figure 1 sensors-19-02673-f001:**
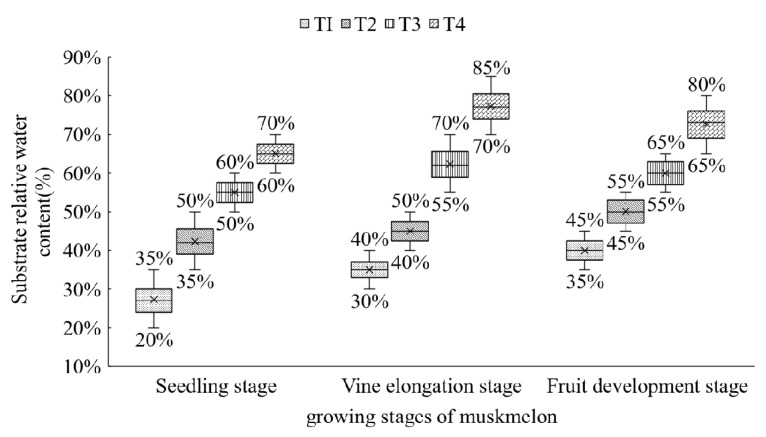
Substrate relative water content (%) in four treatments at three growing stages of muskmelon. T1: 20%~35% relative water content (RWC) at seedling stage, 30%~40% RWC at vine elongation stage, and 35%~45% RWC at fruit development stage. T2: 35%~50% RWC at seedling stage, 40%~50% RWC at vine elongation stage, and 45%~55% RWC at fruit development stage. T3: 50%~60% RWC at seedling stage, 55%~70% RWC at vine elongation stage, and 55%~65% RWC at fruit development stage. T4: 60%~70% RWC at seedling stage, 70%~85% RWC at vine elongation stage, and 65%~80% RWC at fruit development stage.

**Figure 2 sensors-19-02673-f002:**
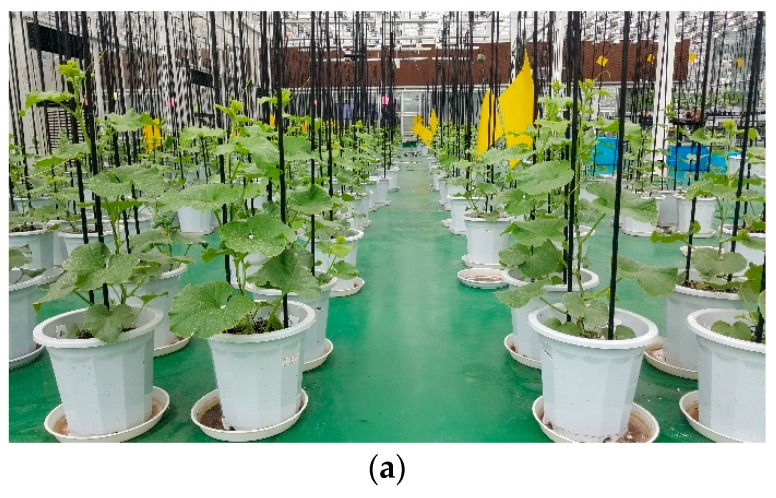
(**a**) Test site of plant cultivation, (**b**) automatic plant conveyors system, (**c**) Lemnatec 3D phenotyping system.

**Figure 3 sensors-19-02673-f003:**
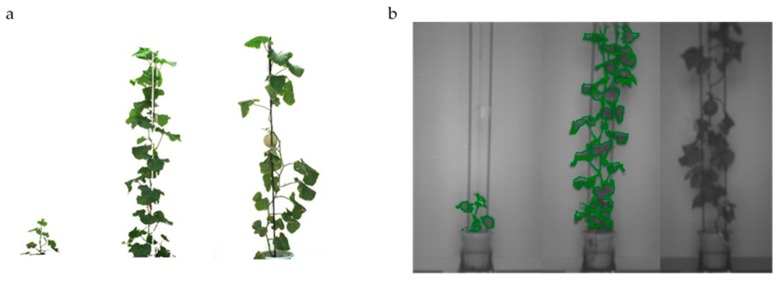
Muskmelon images acquired by Scanalyzer 3D (Lemna, Würselen, Germany) within visible (**a**) and near-infrared (**b**) spectra. From left to right of each panel were the images at seedling stage, vine elongation stage, and fruit development stage, respectively.

**Figure 4 sensors-19-02673-f004:**
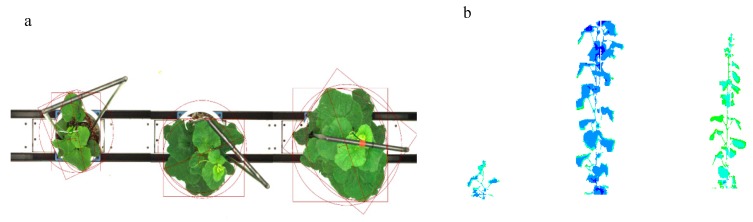
Phenotypic image processing steps. (**a**) Images within visible spectra;(**b**) Images within near-infrared spectra. Different colors represent different water contents in plants.

**Figure 5 sensors-19-02673-f005:**
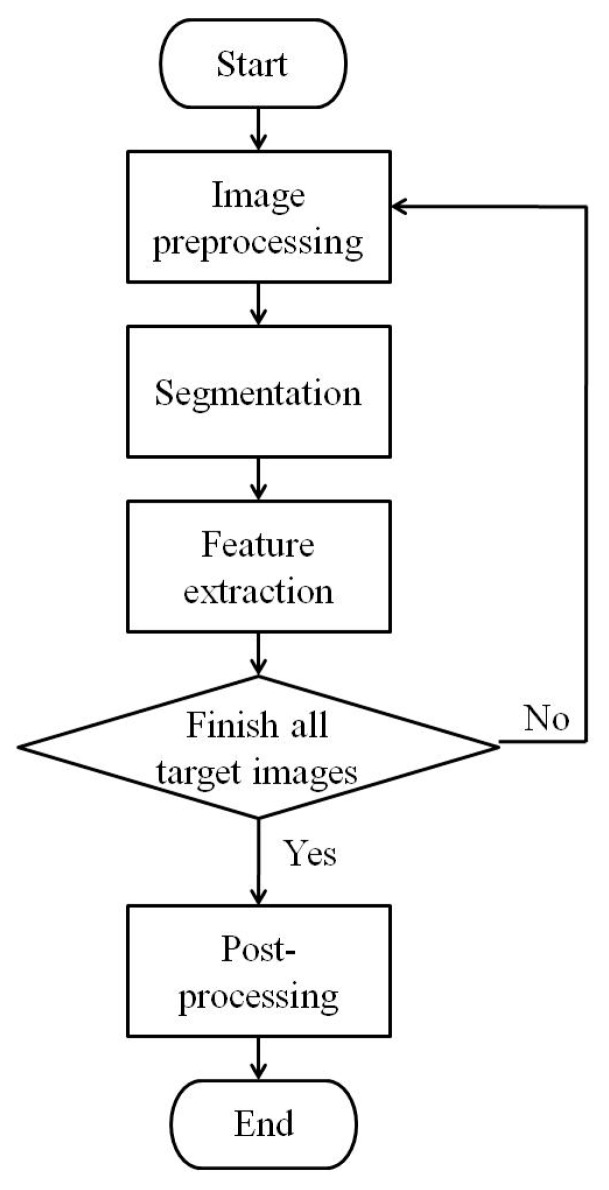
Flowchart of image processing procedure.

**Table 1 sensors-19-02673-t001:** Phenotypic traits extracted from collected images of muskmelon.

Trait	No.	Description	Difference among Four Water Treatments ^1^		
			Seedling Stage	Vine Elongation Stage	Fruit Development Stage
Morphology					
Projection area	1	Area of plant projection (mm^2^)	^*^	^*^	^*^
Object Extent X	2	Minimum width of plant projection (mm)	^*^	^*^	^*^
Object Extent Y	3	Minimum height of plant projection (mm)	^*^	^*^	^*^
DBX area	4	Minimum external polygonal area of plant projection (mm^2^)	^*^	^*^	^*^
Convex hull circumference	5	mm	^*^	^*^	^*^
Diameter	6	Circumscribed circle diameter of plant projection (mm)	NS	^*^	^*^
Min rectangle area	7	Minimum external rectangle area of plant projection (mm^2^)	^*^	^*^	^*^
Compactness	8	Square of the object perimeter to object area	NS	^*^	^*^
Eccentricity	9	The ratio of the distance between the foci to the length of the major axis	NS	^*^	^*^
Invisible light					
Mean blue index	10	Mean blue index of muskmelon image within RGB color space	NS	^*^	^*^
Variance of mean blue index	11	Variance of mean blue index of muskmelon image within RGB color space	NS	^*^	^*^
Mean green index	12	Mean green index of muskmelon image within RGB color space	^*^	^*^	NS
Variance of mean blue index	13	Variance of mean green index of muskmelon image within RGB color space	^*^	^*^	^*^
Mean red index	14	Mean red index of muskmelon image within RGB color space	^*^	^*^	^*^
Variance of mean red index	15	Variance of mean red index of muskmelon image within RGB color space	NS	^*^	^*^
Near-infrared (NIR)					
A1	16	Number of plant pixels with NIR intensity in 122–138	NS	^*^	^*^
R1	17	Ratio of plant pixels with NIR intensity in 122–138	NS	^*^	^*^
A2	18	Number of plant pixels with NIR intensity in 154–170	^*^	^*^	^*^
R2	19	Ratio of plant pixels with NIR intensity in 154–170	^*^	^*^	^*^
A3	20	Number of plant pixels with NIR intensity in 170–186	^*^	^*^	^*^
R3	21	Ratio of plant pixels with NIR intensity in 170–186	NS	^*^	^*^
A4	22	Number of plant pixels with NIR intensity in 186–202	^*^	^*^	^*^
R4	23	Ratio of plant pixels with NIR intensity in 186–202	^*^	^*^	^*^
A5	24	Number of plant pixels with NIR intensity in 202–218	^*^	^*^	^*^
R5	25	Ratio of plant pixels with NIR intensity in 202–218	^*^	^*^	^*^
A6	26	Number of plant pixels with NIR intensity in 218–234	^*^	^*^	^*^
R6	27	Ratio of plant pixels with NIR intensity in 218–234	^*^	^*^	^*^
A7	28	Number of plant pixels with NIR intensity in 234–250	^*^	^*^	^*^
R7	29	Ratio of plant pixels with NIR intensity in 234–250	^*^	^*^	NS

^1^, the underlined *, ^*^, indicates the trait used in simplified random forest model. ”NS” means no significant diffrences.

**Table 2 sensors-19-02673-t002:** Random forest model parameters at the three growing stages of muskmelon.

Stage	Random Forest Model							
	Original				Simplified			
	Mtry	Accuracy (%)	Kappa	Time-Consuming (s)	Mtry	Accuracy (%)	Kappa	Time-Consuming (s)
Seedling stage	2	78.50	0.710	31.08	2	77.60	0.698	12.88
Vine elongation stage	5	95.70	0.941	136.16	5	94.40	0.922	17.44
Fruit development stage	5	99.50	0.993	30.46	1	90.00	0.846	11.14

**Table 3 sensors-19-02673-t003:** Forecasting accuracy of substrate water content using Confusionmatrix.train function in random forest model at three growing stages of muskmelon.

Stage	Random Forest Model							
	Original				Simplified			
Seedling stage								
T1	20.5	1.2	0.6	1.2	19	1.5	1.8	2.7
T2	2.1	16.4	2.4	1.2	1.5	15.8	2.7	2.4
T3	2.4	0.3	25.9	4.2	2.4	0.3	25.9	4.2
T4	0.0	2.7	3.3	15.8	0.0	2.7	3.3	15.8
Mean	0.785				0.776			
Vine elongation stage								
T1	9.8	0.8	0.0	0.0	9.8	1.0	0.0	0.0
T2	0.3	24.6	0.4	0.0	0.3	24.0	1.1	0.2
T3	0.7	0.7	29.6	0.5	0.7	0.7	28.8	0.5
T4	0.0	0.5	0.3	31.8	0.0	0.7	0.5	31.6
Mean	0.957				0.944			
Fruit development stage								
T1	-	-	-	-	-	-	-	-
T2	-	24.0	0.5	0.0	-	17.6	3.7	0.0
T3	-	0.0	40.6	0.0	-	6.4	37.4	0.0
T4	-	0.0	0.0	34.9	-	0.0	0.0	34.9
Mean	0.995				0.900			

-, not fruit set on the plant due to drought. Data were mean value of forecasting accuracy using three 10-fold crosscheck validation.

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
