# Peer review of "A Phenotype-Based Approach for the Substrate Water Status Forecast of Greenhouse Netted Muskmelon"

_sensors, 2019, doi:10.3390/s19122673_

Round 1

Reviewer 1 Report

The authors have satisfactorily responded to all my questions and made the necessary changes to the manuscript. It looks ready for publication as far as I can tell.

Author Response

Dear reviewer,

Thank you very much for your kind comments and great efforts on our manuscript.We also appreciate your professional guidance and valuable suggestions!

Best regards.

Sincerely,

Yilu Yin

Reviewer 2 Report

The language and writing is good. The flow of the paper is clear and logic. Need minor revision to address the following comments.

Please add more information in materials and methods to clarify these points: (1) L99 – L104 Was the imaging only conducted at three days (one in each targeted growth stage)? (2) Random Forest modeling – Please state explicitly how many samples are there in total, among which how many were in the calibration set and how many were in the validation set? I understand your 40 plants were 4 (RWC levels) x 2 (varieties) x 5 (biological replicates). Did you use all 40 for model calibration and reported cross validation results or you split to a calibration/validation set.  Be aware that if only cross-validation is done, your result might be overly optimistic.

Limitation of the study L281-282: I wonder muskmelon was normally grown in the field or in the pot like in this study?  If in the field, I doubt this study would have limited practical implication. In the field, it is not only lighting consistency. Occlusion by other plants is another big problem. Second, there are articles that show NIR or hyperspectral imaging can directly predict leaf water content. See these papers: https://doi.org/10.1016/j.molp.2015.06.005; doi: 10.3389/fpls.2017.01348. The authors should become familiar with these papers. It makes more sense to use imaging to predict plant (water) stress than estimate soil water content. This is a limitation that should be discussed and future study the authors should focus on.

Specific comments: Line 48: A large number of plants. Line 76: China is in East Hemisphere?

Author Response

Dear reviewer,

Thank you for your review of our manuscript (487273).We will provide a point-by-point response to your valuable comments and suggestions.

Point 1: L99 – L104 Was the imaging only conducted at three days (one in each targeted growth stage)?

Response 1: Thank you for asking this question. Imaging was performed every three days during the whole growth period. This has been supplemented in the manuscript.

Point 2: Random Forest modeling – Please state explicitly how many samples are there in total, among which how many were in the calibration set and how many were in the validation set? I understand your 40 plants were 4 (RWC levels) x 2 (varieties) x 5 (biological replicates). Did you use all 40 for model calibration and reported cross validation results or you split to a calibration/validation set. Be aware that if only cross-validation is done, your result might be overly optimistic.

Response 2: Thank you for asking this question. The total samples of experiment from April to July were selected as the training dataset via stratified random sampling method for model development and optimization and the repetitive experiments as the test dataset.

Point 3: Limitation of the study L281-282: I wonder muskmelon was normally grown in the field or in the pot like in this study?  If in the field, I doubt this study would have limited practical implication. In the field, it is not only lighting consistency. Occlusion by other plants is another big problem. Second, there are articles that show NIR or hyperspectral imaging can directly predict leaf water content. See these papers: https://doi.org/10.1016/j.molp.2015.06.005; doi: 10.3389/fpls.2017.01348. The authors should become familiar with these papers. It makes more sense to use imaging to predict plant (water) stress than estimate soil water content. This is a limitation that should be discussed and future study the authors should focus on.

Response 3: Thank you for asking this question. In modern glass greenhouse cultivation, soilless vertical cultivation is often used, which can increase the number of plants planted in limited greenhouse area, improve land use efficiency, facilitate management and mechanized operation, and avoid the problem of shading under reasonable plant spacing. Therefore, it is of practical significance for modern facility planting.We are familiar with these papers(https://doi.org/10.1016/j.molp.2015.06.005; doi: 10.3389/fpls.2017.01348) and have cited them. Next step, we will study a variety of optical sensor fusion technologies, such as near infrared and hyperspectral, to predict plant water.

Point 4: Specific comments: Line 48: A large number of plants. Line 76: China is in East Hemisphere?

Response 4: Thank you for your advice on the details of this article. We have revised the sentence for Line 48, Moreover, with the development of plant phenotypic monitoring technologies, now it is viable to carry out automatic high-throughput monitoring without damaging the plants, thus quickly acquiring the phenotypic traits and remarkably shortening the test cycle. We have revised the sentence for Line 76, the experiment was conducted in the greenhouse of Shanghai Zealquest Technology Co., Ltd, China (GPS coordinate: 31°11′N, 121°36′E) from April to July in 2016. The repetitive experiments was conducted in a multiplan greenhouse at Shanghai Jiaotong University, China (31°11´ N, 121°36´E) from August to November in 2016, the glasshouse heating, ventilation, internal and external shading were all automatically regulated by Priva Software (Priva Company, The Netherlands).

Thanks again for your careful work and  valuable suggestions.
Best regards.

Sincerely,

Liying Chang

This manuscript is a resubmission of an earlier submission. The following is a list of the peer review reports and author responses from that submission.

Round 1

Reviewer 1 Report

Manuscript: ‘A Phenotype Based Approach for the Substrate Water Status Forecast of Greenhouse Netted Muskmelon’

In this work a model has been used to forecast plant substrate water status given the phenotypic traits throughout the muskmelon (varieties: ‘Wanglu’ and ‘Arus’) growing season. They were planted in a greenhouse under four substrate water treatments and their phenotypic traits were measured by taking images within the visible and near-infrared spectrums, respectively. Unlike other similar works, the Random Forest package of R language was used for modeling. The study addresses to a quite interesting topic, in term of agricultural applications; therefore I think that the results of this study could contribute to improve management strategies of irrigation in crop production (in this case: netted muskmelon). Furthermore, this is a relevant topic lies within the scope of the MDPI sensors journal. The article is well organized and neatly written with the appropriate scientific content. Based on the above, I support the publication of this manuscript, but only after a minor revision.

********************************

Title: it fits perfectly the paper content. 

Abstract: it is quite adjusted to the paper content.

Introduction: this section provides sufficient background and includes relevant references about the importance of the substrate water content throughout all growing stages of netted muskmelon and the machine learning algorithms. Objectives clearly stated.

Materials and Methods: the study unit, materials and treatments have been clearly stated. In my opinion, the research shows a design appropriated and its methods have been adequately described, but for the sake of clarity, I think that this section could be significantly improved if the authors add a flow chart with the different methods described in text, highlighting inputs, applied analysis/procedure, and outputs so that readers could understand this section easier.  Nevertheless, I have some specific comments: 

Line 76: what is the type of soil and field water capacity in these 40 potted plants? Please, indicate (e.g., soil-water content averaged).

Lines 88-89: authors should indicate those tools used for preprocessing, image segmentation, and traits extraction (i.e., did you use R scripts? clarify). 

Results: the results have been presented clearly, but I have a specific comment:

Table 1: authors could replace this table with boxplots in order to provide more information to the reader (e.g., median, dispersion, quartiles and extremes)

Discussion:  this section is clear and concise and is supported by relevant references.

Conclusions: these are clear and concise and are supported by the results.

Reviewer 2 Report

This work present a Random Forest application for assessing substrate water content through plant phenotypic traits. While the subject is interesting, the authors need to make greater effort in the presentation but also the substance of their work. Starting from the introduction as well as the number of works cited, it is obvious that the authors have not researched the state of the art adequately. This also impacts their discussion.